# LongVPO: From Anchored Cues to Self-Reasoning for Long-Form Video Preference Optimization

**Zhenpeng Huang**[1,*]     **Jiaqi Li**[2,*]     **Zihan Jia**[1]     **Xinhao Li**[1,3]     **Desen Meng**[1]
**Lingxue Song**[2]     **Xi Chen**[2]     **Liang Li**[2]     **Limin Wang**[1,3,†]
[1]State Key Laboratory for Novel Software Technology, Nanjing University
[2]JIUTIAN Research     [3]Shanghai AI Laboratory
https://github.com/MCG-NJU/LongVPO

## Abstract

We present LongVPO, a novel two-stage Direct Preference Optimization framework that enables short-context vision-language models to robustly understand ultra-long videos without any long-video annotations. In Stage 1, we synthesize preference triples by anchoring questions to individual short clips, interleaving them with distractors, and applying visual-similarity and question-specificity filtering to mitigate positional bias and ensure unambiguous supervision. We also approximate the reference model's scoring over long contexts by evaluating only the anchor clip, reducing computational overhead. In Stage 2, we employ a recursive captioning pipeline on long videos to generate scene-level metadata, then use a large language model to craft multi-segment reasoning queries and dispreferred responses, aligning the model's preferences through multi-segment reasoning tasks. With only 16K synthetic examples and no costly human labels, LongVPO outperforms the state-of-the-art open-source models on multiple long-video benchmarks, while maintaining strong short-video performance (e.g., on MVBench), offering a scalable paradigm for efficient long-form video understanding.

## 1   Introduction

Recent vision-language models (VLMs)[32, 8, 60, 7, 41, 25] have demonstrated impressive capabilities in both image and video understanding. However, their performance often degrades when applied to tasks that require long-context visual reasoning, such as analyzing videos that span over an hour [48, 14, 62]. This presents a significant challenge in scaling VLMs for long-form video understanding.

While recent progress in long-video VLMs [43, 40, 7] has been encouraging, most approaches rely heavily on costly, high-quality annotations for long videos, limiting their scalability in practical applications. In contrast, existing short-context VLMs—despite being trained only on limited-frame inputs—have shown surprisingly competitive results on long-video benchmarks, largely thanks to their strong foundational vision-language alignment. This observation suggests a promising direction: short-context VLMs may possess untapped potential for long-video modeling if properly extended. This raises a natural question: *How far can we push short-context VLMs into the long-video regime—without the burden of expensive re-training or labels?*

To explore this question, we start with a strong short-context VLM [8, 19] that was not trained with long-range visual inputs and evaluate its performance on long-video understanding tasks. We identify two key challenges that limit its effectiveness: **(1) Scarcity of Long-Form Video Annotations:**

---

*Equal contribution.
†Corresponding author (lmwang@nju.edu.cn).

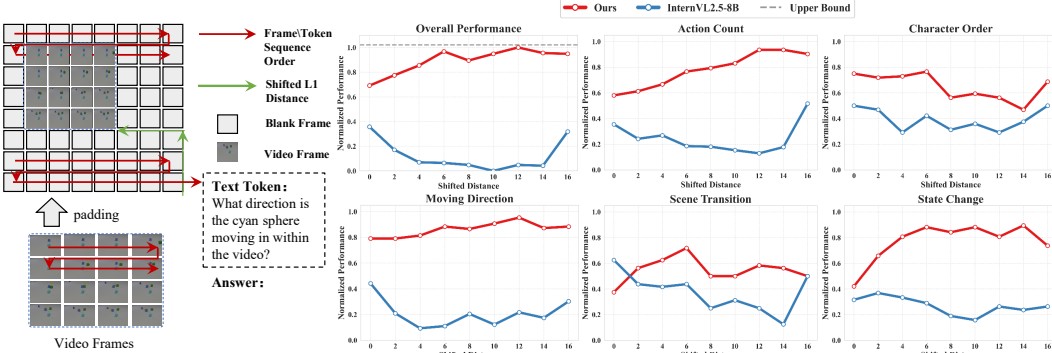

Figure 1: **Context Position Bias Probing. Left:** A short video segment (visualized as a $4 \times 4$ grid) is embedded within a much longer padded sequence and processed chronologically. **Right:** Performance is plotted against each frame's L1 distance from the question token. The middle-position drop indicates a strong positional bias ("lost-in-the-middle"). The Upper Bound shows performance without padding, revealing degradation under long-context settings.

High-quality video-text annotations, such as detailed captions or question-answer pairs, are typically available only for short clips, where annotators can reasonably cover the content. For long videos spanning tens of thousands of frames, such annotation becomes prohibitively expensive [54] and often suffers from incomplete coverage and poor temporal alignment [47]. **(2) Context-Length Bias in Short-Context VLMs:** Common practices such as YARN [35], NTK can extend positional encodings for longer sequences, but the resulting models still suffer from position-related biases and limited performance gains. To simulate long-context scenarios, we embed the original short video's 1D frame sequence within a much longer sequence, padding the surrounding positions with blank frames (visualized as a grid in Fig. 1). This setup allows us to investigate the model's sensitivity to spatial positions across extended contexts. Specifically, by computing the L1 distance between each frame's position and a fixed query point, we uncover a "lost-in-the-middle" phenomenon—analogous to what has been observed in long-sequence language models [3]—where the model's performance dips for inputs located near the center of the grid. This highlights a positional bias that disfavors centrally located content (see Fig. 1).

Recent efforts [23] attempt to address this by leveraging Direct Preference Optimization (DPO) to enhance grounding capabilities. However, as depicted in Fig. 2, this method assumes access to a reference model that already supports long-context reasoning. This assumption does not hold for short-context VLMs. Moreover, this approach requires proprietary models to generate and filter preference data, which introduces external language model biases without fully viewing the video. As a result, the method fails to fundamentally resolve the problem and delivers suboptimal performance.

To address these challenges, we propose a two-stage training framework that extends short-context VLMs to ultra-long video contexts, as shown in Fig. 3: **Stage 1: Efficient Short-to-Long Learning from Anchored Cues.** We form mixed, interleaved sequences of short clips from the SFT dataset. For each clip, we generate an anchor question and use the short-context VLM's answer as the Preferred Response, ensuring via scalable auto-filtering that each question refers to exactly one clip. The model learns to maximize the likelihood of the Preferred Response given the anchor question and its corresponding clip. To simulate distracting contexts, we introduce Dis-Preferred Responses by prompting temporally misaligned clips, forcing the model to retrieve the correct answer from many candidates. We also randomize the target clip's position within the sequence to mitigate positional biases during training. **Stage 2: Self-Training for Long Video Preference Alignment.** Building on Stage 1's memory and retrieval single-segment skills, we train the model to handle longer, more complex videos, without requiring ground-truth annotations. First, we employ a recursive captioning pipeline to generate structured, scene-level metadata to leverage the model's short-context capabilities. We then transfer insights from open-source LLMs' long-text understanding: given a sequence of scene-organized captions, we generate questions and identify the minimal set of scenes needed to answer them. We then craft Dis-Preferred Responses by prompting the model with partial or misleading context (e.g., omitting critical scenes), encouraging it to assemble the complete information chain required for accurate answers in real-world, long-video scenarios.

In summary, our contributions are threefold:

- We introduce LongVPO, a two-stage framework that extends short-context VLMs to long video contexts without relying on any long-video annotations.

- We construct a synthetic DPO training set from short visual context transfer to long text context for long videos, using only ~16k instances—significantly fewer than existing instruction-tuning datasets —eliminating the need for long-video labels.

- Our approach outperforms existing long-video models trained on large-scale supervised and preference-optimized data across challenging long-video understanding benchmarks, while maintaining competitive performance on short-video tasks, offering a new superior paradigm for efficient multimodal long video understanding.

## 2 Related work

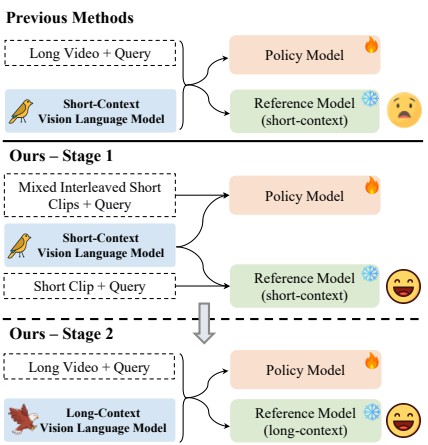

Figure 2: Comparison of prior methods with our proposed two-stage method.

**VLMs for Long Video Understanding.** Advancements in vision-large models (VLMs) [8, 11, 21, 20, 26, 31, 60] have shown impressive capabilities in video understanding, and many models demonstrated excellent performance in short video analysis. Some recent works make further efforts towards long video understanding by designing certain strategies to compress/select visual context [10, 13, 17, 22, 40, 42] or extend the temporal context window [7, 41, 57]. In addition to the innovation in model architecture, it is also important to construct long-form video instruction datasets to guide VLMs in extracting detailed visual cues and modeling cross-temporal relationships. Representative methods such as VideoChat-Flash [25], Kangaroo [27], and Video-XL [42] build data production pipelines and curate long video data to enhance the ability of video VLM. However, high-quality long video annotations can be expensive and time-consuming, making them difficult to obtain and scale up compared to short videos [6]. Therefore, developing an efficient strategy to employ short video-text data to facilitate VLMs in long video understanding remains a challenge. In this work, we propose a novel two-stage framework for VLMs to progressively learn the ability to analyze longer videos with short video annotations.

**DPO for Video-VLMs.** As a post-training strategy, DPO has been frequently adopted in the development of VLMs [15, 28, 49, 61, 63]. Unlike the next-token prediction used in the SFT step, DPO refines the VLM using triplets of queries, preferred and rejected responses, reducing model hallucination and better aligning with human reasoning [16]. The simplicity and strong performance of DPO training further encourage researchers to apply it to video-based VLMs, where devising effective spatial and temporal perturbation tasks is a crucial part [2, 16, 18, 23, 24, 55, 58]. Recent works propose methods such as frame cutout, spatial misalignment, clip dropping, clip rearrangement, and frame disconnection to generate query and corresponding preference responses, with the help of proprietary or open-sourced models [16, 23, 24, 55]. The curated data are employed in the VLM training to enhance the spatial-temporal perception and dynamic modeling capabilities for videos. While these efforts have shown promise, many existing works focus on minute-level or short-form videos. While recent efforts [39, 5, 44] have shown promise in long text context alignment for LLMs, extending this paradigm to video presents distinct challenges. For long visual context, directly generating preference data remains challenging due to task design complexity and high computation costs, a limitation even in recent explorations of DPO for long videos [23, 24]. Therefore, our progressive DPO training that incrementally extends the model's capacity to capture long temporal dependencies may offer a more practical and scalable path towards long video understanding.

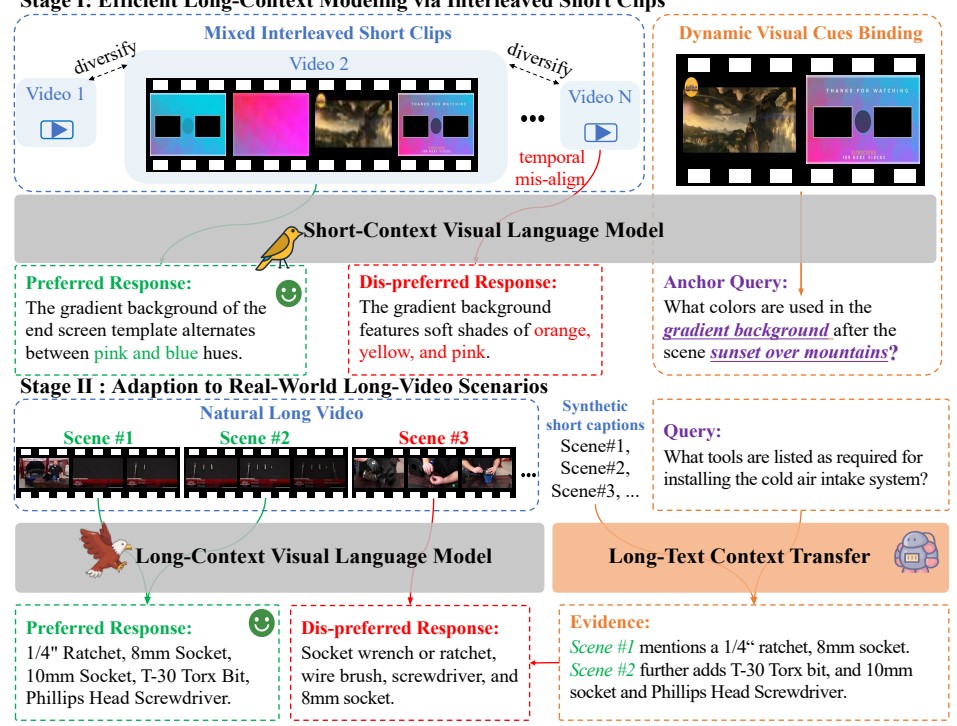

Figure 3: Overview of our two-stage training framework. **Stage 1:** Short clips are concatenated to form a pseudo-long video. The target model generates the query ($q_i$) and preferred response ($y_i^+$) conditioned on the *anchor* clip and its caption, while dispreferred responses ($y_i^-$) are generated by prompting the model to answer based on non-anchor (distractor) clips, simulating temporal misalignment errors. **Stage 2:** For unlabeled long videos, an LLM generates the query ($q_i$) and reasoning trace ($r_i$) based on scene synthetic captions. The *target MLLM* then generates the preferred response ($y_i^+$) based on corresponding scene context, and generates dispreferred responses ($y_i^-$) under degraded context (e.g., partial or irrelevant scenes).

# 3 Method

## 3.1 Background

Direct Preference Optimization (DPO) [36] aligns a policy model $\pi_\theta$ with human preferences by directly optimizing a policy that best satisfies the preferences, using a simple classification loss. The objective function is formulated as maximizing the log-sigmoid of the log-likelihood ratio between preferred ($y_i^+$) and dispreferred ($y_i^-$) responses, relative to a frozen reference model $\pi_{\text{ref}}$:

$$\mathcal{L}_{\text{DPO}}(\theta) = -\sum_i \log \sigma \left[ \beta \left( \log \frac{\pi_\theta(y_i^+ \mid x_i)}{\pi_{\text{ref}}(y_i^+ \mid x_i)} - \log \frac{\pi_\theta(y_i^- \mid x_i)}{\pi_{\text{ref}}(y_i^- \mid x_i)} \right) \right], \tag{1}$$

where $\sigma$ denotes the sigmoid function, and $\beta > 0$ is a hyperparameter that controls the strength of the preference margin, effectively determining how sharply the policy should prefer $y_i^+$ over $y_i^-$.

## 3.2 Distribution Shift from Short to Long Video Contexts

Consequently, directly applying a model fine-tuned on short-video data to preference optimization tasks in long videos reveals two critical challenges:

- **Reference-model degradation in extended contexts:** As depicted in Fig. 1, the short-context visual model not only exhibits position bias but also shows over-specialization to short temporal spans, limiting its generalization to extended video contexts. Therefore, in the DPO framework, the reference model $\pi_{\text{ref}}$ is typically kept frozen. As the policy model

$\pi_\theta$ is trained to understand and leverage long-range dependencies inherent in extended video sequences, its output characteristics for queries over these long videos may diverge substantially from the short-video patterns that $\pi_{\text{ref}}$ was trained to evaluate. This divergence can impair the ability of $\pi_{\text{ref}}$ to serve as a consistent and meaningful baseline in the DPO objective, potentially leading to unstable or suboptimal policy updates when adapting to long-video tasks.

- **Annotation scarcity and distribution gap:** The lack of dense, high-quality annotations for long video sequences creates a significant distributional disparity between the data available for initial supervised fine-tuning (SFT) or reference model training (typically short clips) and the target domain of long videos. This data gap can severely degrade model performance when generalizing to longer contexts.

### 3.3 Stage 1: Efficient Short-to-Long Learning from Anchored Cues

To mitigate the distribution shift from short to long video contexts and effectively leverage abundant short-video SFT data without compromising performance on short clips, we design an anchor-based approach that preserves short-context fidelity while exposing the model to long-range contextual variation. This strategy encompasses three core components: the synthesis of anchor-centric preference triples, their subsequent refinement through filtering, and a specific adaptation of the DPO objective to robustly incorporate this data.

The synthesis of preference data forms the initial component. At its core is the generation of preference triples $(q_i, y_i^+, y_i^-)$ where the query $q_i$ is answerable only from a designated "anchor" short clip within a longer composite video $x_i$. This *Dynamic Visual Cues Binding* process involves three key steps:

- **Anchor-centric QA and Preferred Response Generation:** A short video clip, potentially with supplementary annotations (e.g., captions), is randomly selected from SFT data to serve as the anchor $x_{i,\text{anchor}}$. A question-answer (QA) pair $(q_i, y_i^+)$ is then generated by the target short VLM such that $q_i$ can only be answered comprehensively using information present exclusively within $x_{i,\text{anchor}}$. The answer $y_i^+$ constitutes the preferred response.
- **Composite Sequence Assembly:** Multiple distinct short clips, including the designated anchor $x_{i,\text{anchor}}$, are concatenated to form the longer composite video sequence $x_i = [x_{i,1}, ..., x_{i,\text{anchor}}, ..., x_{i,k}]$.
- **Plausible Dispreferred Response Generation:** For the same question $q_i$, a dispreferred response $y_i^-$ is generated. This response is designed to be plausible yet incorrect by drawing information from non-anchor clips within $x_i$ to simulate anchor positioning errors.

Following data synthesis, a critical step is to ensure the quality and unambiguity of the preference signals. To guarantee that the anchor clip $x_{i,\text{anchor}}$ is genuinely unique in containing the necessary information to answer $q_i$, we introduce two complementary post-filtering mechanisms:

- **Scene-Similarity Filtering:** Any non-anchor clips within $x_i$ whose vision embedding (e.g., DINOv2 [33]) similarity to the anchor $x_{i,\text{anchor}}$ exceeds a predefined threshold are replaced, or the entire sample is discarded. This ensures greater visual dissimilarity between the anchor and distractor segments.
- **Question Specificity Filtering:** An Optional LLM (e.g., Qwen-2.5 32B [50]) verifies that $q_i$ depends on multiple distinct visual elements within $x_{i,\text{anchor}}$. Questions lacking this specificity, which could be answered by other clips, are discarded.

As discussed previously, applying the short-clip reference model $\pi_{\text{ref}}$ to the full input $x_i$ leads to performance degradation due to context-length mismatch. To address this, we introduce an **anchor-only approximation**, which leverages the design hypothesis that only the anchor clip $x_{i,\text{anchor}}$ contains information necessary to answer $q_i$, while non-anchor segments provide no relevant signal. This hypothesis is supported by our filtering process, which reduces semantic similarity between anchor and non-anchor clips, reinforcing the anchor's informational sufficiency.

Under this approximation, the reference model's likelihood is evaluated solely on the anchor clip:

$$\pi_{\text{ref}}(y \mid x_i) \approx \pi_{\text{ref}}(y \mid x_{i,\text{anchor}}). \tag{2}$$

This avoids context-length mismatch, reduces computational and memory costs, and ensures likelihoods reflect only anchor-related content. The modified DPO objective thus becomes:

$$\mathcal{L}_{\text{stage1}}(\theta) = -\sum_i \log \sigma \left[ \beta \left( \log \frac{\pi_\theta(y_i^+ \mid x_i)}{\pi_{\text{ref}}(y_i^+ \mid x_{i,\text{anchor}})} - \log \frac{\pi_\theta(y_i^- \mid x_i)}{\pi_{\text{ref}}(y_i^- \mid x_{i,\text{anchor}})} \right) \right]. \quad (3)$$

### 3.4 Stage 2: Self-Training for Long Video Preference Alignment

While the efficient Short-to-Long Learning method scales input length via synthetic clip compositions, such sequences often lack the natural coherence and narrative structure of real long videos, which is crucial for preference learning that depends on temporally grounded reasoning and causal event understanding. This becomes problematic for queries requiring temporal reasoning of multiple cues (e.g., action chains or evolving events). To bridge this gap, we propose a self-training framework for aligning with long-video preferences.

**Data Preparation.** In this stage, we first employ a recursive captioning strategy to generate dense textual descriptions for long videos. For each temporally segmented scene within a long video, the target model is conditioned on both the current video segment and the captions generated for preceding scenes. This iterative process constructs a coherent, context-aware caption sequence for the entire video, capturing local semantics and their broader contextual dependencies.

**Construction of Preference Data** $(q_i, y_i^+, y_i^-)$ **for Self-Training.** The generation of preference triples involves a multi-step process, leveraging an LLM for query generation and the target MLLM for response generation.

1. **Long-Context Knowledge Transfer.** We prompt an LLM to process the video content (represented by scene IDs and captions) to produce a pair $(q_i, r_i)$. Here, $q_i$ is a video-related query, and $r_i$ is a detailed reasoning trace. Crucially, $r_i$ must explicitly cite specific scene IDs (e.g., "Scene #N"), which serve as **binary scene-question relevance labels**.

2. **Preferred Response $(y_i^+)$ Generation.** For each query $q_i$ and the corresponding full long video $x_i$, the target MLLM $\pi_\theta$ is prompted to generate a response. This output, denoted as $\pi_\theta(y \mid q_i, x_i)$, serves as the preferred response $y_i^+$. This step utilizes the model's capability to reason over the full visual context to articulate a comprehensive answer.

3. **Dispreferred Response $(y_i^-)$ Generation.** Dispreferred responses $y_i^-$ are generated by *degrading the visual context* provided to the target MLLM. We employ two strategies to induce plausible but flawed responses: *Reasoning from Partial Evidence*, where the MLLM is prompted with only a subset of the relevant scenes identified in $r_i$ to force incomplete reasoning; and *Hallucination from Irrelevance*, where the model is prompted using only non-relevant scenes (those not cited in $r_i$). The resulting outputs, which typically hallucinate details or miss critical evidence, serve as the dispreferred responses $y_i^-$.

For Stage 2, we employ the standard DPO objective $\mathcal{L}_{\text{stage}_2}(\theta) = \mathcal{L}_{\text{DPO}}(\theta)$. While the self-generated $y_i^+$ may not be perfect, the relative preference delta $(y_i^+, y_i^-)$ provides a valid training signal. The policy model $\pi_\theta$ is initialized from the Stage 1 checkpoint, and the reference model $\pi_{ref}$ is frozen as the Stage 1 checkpoint, which has been equipped with the basic capability to retrieve query-relevant clips from the full long-video input.

### 3.5 Total Objective

In both stages $i = 1, 2$, we incorporate the SFT loss into the DPO framework, weighted by $\alpha$ following [34]. The total objective is defined as:

$$\mathcal{L}(\theta) = \mathcal{L}_{\text{stage}_i}(\theta) + \alpha \cdot \frac{-\log \pi_\theta(y^+ \mid x)}{|y^+|}, \quad (4)$$

where $\mathcal{L}_{\text{stage}_i}(\theta)$ denotes the DPO loss at stage $i$, $\pi_\theta(y^+ \mid x)$ represents the model likelihood of the preferred response $y^+$.

| Models | Size | Frames | LVBench Overall | LongVideoBench Validation | MLVU M-Avg | Video-MME (wo / w sub) Overall | Long | MVBench Overall |
|---|---|---|---|---|---|---|---|---|
| **Average Duration** | | | 4101s | 473s | 651s | 1,010s | 2,386s | 16s |
| *Proprietary Vision-Language Models* | | | | | | | | |
| GPT4-V [1] | Undisclosed | - | 59.1 | 49.2 | 59.9 / 63.3 | 53.5 / 56.9 | 43.7 |
| GPT4-o [32] | Undisclosed | 30.8 | 66.7 | 64.6 | 71.9 / 77.2 | 65.3 / 72.1 | 64.6 |
| Gemini-1.5-Pro [38] | Undisclosed | 33.1 | 64.0 | - | 75.0 / 81.3 | 67.4 / 77.4 | 60.5 |
| *Open-Source Multi-Image Vision-Language Models* | | | | | | | | |
| LLaVA-OneVision [19] | 72B | 32 | - | 61.3 | 66.4 | 66.3 / 69.6 | 60.0 / 62.4 | 59.4 |
| InternVL2 [9] | 76B | 16 | - | 61.0 | 69.9 | 61.2 / 67.8 | - | 69.6 |
| LLaVA-OneVision [19] | 7B | 32 | - | 56.5 | 64.7 | 58.2 / - | - | 56.7 |
| Oryx-1.5 [29] | 7B | 128 | - | 56.3 | 67.5 | 58.8 / 64.2 | - | - |
| MiniCPM-v2.6 [52] | 8B | 64 | - | 54.9 | 37.3 | 60.9 / 63.7 | 51.8 / 56.3 | - |
| mPLUG-Owl3 [53] | 7B | 16 | - | 52.1 | 63.7 | 59.3 / - | 50.1 / - | - |
| Qwen2-VL [50] | 7B | 2FPS | - | 55.6 | - | 63.3 / 69.0 | - | 67.0 |
| NVILA [30] | 7B | 256 | - | - | 70.1 | 64.2 / 70.0 | 54.8 / 63.3 | - |
| *Open-Source Video-Language Models* | | | | | | | | |
| VideoLLaMA2 [11] | 72B | 16 | - | - | 61.2 | 62.4 / 64.7 | 57.6 / 59.0 | - |
| LLaVA-Video [60] | 7B | 1FPS | - | 58.2 | 70.8 | 63.3 / 69.7 | - | 58.6 |
| Video-XL [42] | 7B | 2,048 | - | 49.5 | 64.9 | 55.5 / 61.0 | 49.2 / - | - |
| VideoLLaMA2 [56] | 7B | 16 | - | - | 48.5 | 47.9 / 50.3 | - | - |
| Video-CCAM [13] | 9B | 96 | - | - | 58.5 | 53.2 / 57.4 | 46.7 / 49.9 | - |
| Kangaroo [27] | 8B | 64 | - | 54.8 | 61.0 | 56.0 / 57.6 | 46.7 / 59.3 | 61.0 |
| LongVU [40] | 7B | 1FPS | - | - | 65.4 | 60.6 / 59.5 | - | 66.9 |
| LongVA [57] | 7B | 128 | - | - | 56.3 | 52.6 / 54.3 | 46.2 / 47.6 | - |
| LongVILA [7] | 7B | 256 | - | 57.1 | - | 60.1 / 65.1 | - | 67.1 |
| VideoChat-Flash [25] | 7B | 512 | 48.2 | 64.7 | 74.7 | 65.3 / 69.7 | 55.4 / 63.3 | 74.0 |
| InternVL2.5 [8] | 8B | 64 | 43.2 | 60.0 | 68.9 | 64.2 / 66.9 | - | 72.0 |
| InternVL2.5* [8] | 8B | 512 | 45.2 | 62.7 | 67.6 | 61.1 / 65.3 | 51.1 / 57.2 | 72.0 |
| **+LongVPO (128f)** | | | | | | | | |
| Stage1 | 8B | 512 | **49.4** (+4.2) | **65.4** (+2.7) | **73.5** (+5.9) | **64.2** (+3.1) / **70.1** (+4.8) | **53.8** (+2.7) / **62.8** (+5.6) | **72.9** (+0.9) |
| Stage2 | 8B | 512 | **50.1** (+4.9) | **66.6** (+3.9) | **74.1** (+6.5) | **64.6** (+3.5) / **70.3** (+5.0) | **55.3** (+4.2) / **64.2** (+7.0) | **73.1** (+1.1) |
| **+LongVPO (256f)** | | | | | | | | |
| Stage1 | 8B | 512 | **49.6** (+4.4) | **66.0** (+3.3) | **74.8** (+7.2) | **65.0** (+3.9) / **71.2** (+5.9) | **55.8** (+4.7) / **65.1** (+7.9) | **72.9** (+0.9) |
| InternVideo2.5* [46] | 8B | 512 | 47.4 | 63.2 | 72.8 | 63.3 / 71.1 | 52.6 / 65.1 | 75.7 |
| **+LongVPO (256f)** | | | | | | | | |
| Stage1 | 8B | 512 | **50.9** (+3.5) | **67.0** (+3.8) | **74.0** (+1.2) | **65.4** (+2.1) / **72.6** (+1.5) | **54.3** (+1.7) / **67.0** (+1.9) | 74.7 |
| Stage2-iter1 | 8B | 512 | **51.0** (+3.6) | **67.2** (+4.0) | **74.4** (+1.6) | **65.6** (+2.3) / **72.5** (+1.4) | **54.9** (+2.3) / **67.1** (+2.0) | 75.1 |
| Stage2-iter2 | 8B | 512 | **51.0** (+3.6) | **67.2** (+4.0) | **74.7** (+1.9) | **66.1** (+2.8) / **73.1** (+2.0) | **56.1** (+3.5) / **67.4** (+2.3) | 75.1 |
| **+LongVPO (512f)** | | | | | | | | |
| Stage1 | 8B | 512 | **50.4** (+3.0) | **67.8** (+4.6) | **75.0** (+2.2) | **65.6** (+2.3) / **73.0** (+1.9) | **54.9** (+2.3) / **67.1** (+2.0) | 75.1 |

Table 1: Accuracy (%) on the short and long video understanding benchmarks. **Size** indicates the number of parameters. **Frames** denotes the maximum number of frames sampled from each video or the frame sampling rate (FPS). The best and second-best results among open-source models of similar size (7∼9B) are in **bold** and underlined, respectively. "256f"/"512f" refer to the maximum number of training frames. * denotes reproduced results.

## 4 Experiment

### 4.1 Implementation Details

**Baseline.** We adopt InternVL-2.5-8B [8] as the base model of our framework. It comprises InternViT-300M as the vision encoder and InternLM-2.5-7B-32K [4] as the language backbone. According to the official report, the model was trained on a maximum of approximately 32 video frames, corresponding to a visual context length of around 8192 tokens. We implement DeepSpeed Ulysses sequence parallelism to enable efficient training with 32K extended video context length.

**Data Preparation.** To ensure a fair and leakage-free evaluation, we rely solely on publicly available datasets. Specifically, Stage 1 training utilizes caption annotations from LLaVA-Video-178K [59]. For Stage 2, we incorporate scene-segmented but unlabeled long videos from Vript [51], both of which are included in the InternVL-2.5 SFT dataset. In total, we utilize 16k training samples, consisting of 10k for Stage 1 and 6k for Stage 2.

For Stage 1 data, we preprocess each source clip from LLaVA-Video-178K by uniformly sampling up to 64 frames at 1 fps. To construct each composite training instance $x_i$, we designate one clip as the *anchor video* ($x_{i,\text{anchor}}$) and randomly select several additional clips as *non-anchor video*. Any non-anchor video whose DINO embedding cosine similarity with the anchor exceeds 0.6 is discarded.

For Stage 2 data, leveraging the pre-segmented scenes in Vript, we employ an external LLM (Qwen2.5-32B) to generate complex queries requiring cross-scene reasoning based on scene captions, along with a relevance chain identifying scenes associated with the question. The preferred response

is generated by feeding the model the query and only the critical video scenes determined by the chain. Conversely, the dispreferred response is synthesized by intentionally inducing errors based on the strategies detailed in Sec. 3.4.

**Video Understanding Benchmarks.** To comprehensively evaluate our model's long-context understanding capabilities, we adopt three existing long-video benchmarks [45, 48, 62] along with the comprehensive benchmark VideoMME [14]. Additionally, we verify performance on MVBench [20]. The evaluated video durations from all these benchmarks span from a few seconds to 2 hours.

## 4.2 Main Results

Our main results are shown in Tab. 1. **LongVPO exhibits strong competitiveness among models of comparable scale on long-context video understanding benchmarks.** Notably, most competing long-video models are trained on datasets with manual curation [12, 37] or rely on proprietary MLLMs [32, 38] to annotate. In contrast, our approach leverages only around 16K synthetic samples, without any reliance on expensive human annotations or closed-source tools, underscoring the effectiveness of our training strategy and data construction methodology.

**LongVPO maintains strong performance in short-video analysis.** Although our primary goal is to improve long-video understanding, our model also achieves competitive results on the general-purpose short-video benchmark MVBench [20], even surpassing prior results with a +1.1 improvement. This further illustrates the effectiveness of LongVPO in accommodating videos of varying durations in real-world scenarios.

**Effectiveness of stage 1.** In Stage 1, the model is trained on synthetic long-video data, resembling a form of structureless memory training. The primary goal is to mitigate context bias and activate the model's localization ability. As shown in Tab. 1, the model generalizes well to real-world scenarios and the objectives are largely fulfilled.

**Effectiveness of stage 2.** Instead of Stage 1 focusing on localizing a single segment to answer questions, Stage 2 focuses on aggregating information across multiple segments from real long videos, thereby enhancing the model's capacity to extract complex, question-relevant content. We observe consistent improvements in most settings, indicating that training on real video domains results in better alignment with realistic scenarios.

**Long video context evaluation.** Following the NIAH setup from LongVA [57], we densely sample multiple frames from a long video and insert a selected image at different positions within the sampled frames. We evaluate InternVL2.5 on a maximum of approximately 3k frames. The results in Fig. 4 show that the baseline model begins to exhibit significant performance degradation at around 800 frames, with complete failure to follow instruction output formats when reaching approximately 1k frames, while LongVPO demonstrates superior long-context modeling capabilities.

## 4.3 Extending to Dedicated Long-Context Models

While the primary results are based on short-video models, we further validate the generalizability of our approach by applying it to InternVideo2.5 [46], a representative long-video model. InternVideo2.5 incorporates two key designs for long-context understanding: (1) pre-training on high-quality long-video datasets, supporting up to 256 frames during training; (2) a specialized redundancy compression mechanism in its vision-language connector for efficient long-context processing.

**Generality beyond short-context models.** Although initially designed under short-video assumptions, our method demonstrates strong generalization. When applied to InternVideo2.5 for continued training, LongVPO consistently surpasses its counterparts trained on InternVL2.5. This indicates that InternVideo2.5 has not yet saturated in long-context understanding, and our approach provides further enhancement, underscoring its adaptability even for models pre-trained on long videos.

**Context length scalability in Stage1.** We evaluate the scalability of our approach under varying context lengths. For Stage 1, we extend the length of the synthesized video by simply selecting more clips. Our approach consistently improves performance as the maximum number of input frames increases (e.g., from LongVPO-256f to LongVPO-512f), demonstrating strong scalability and an enhanced ability to exploit longer temporal contexts.

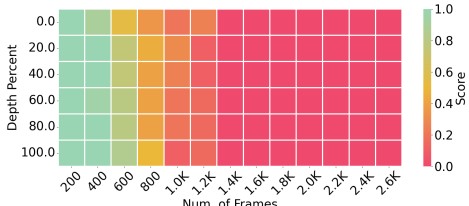
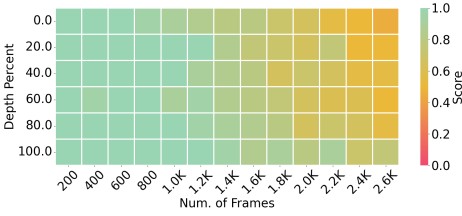

(a) Baseline Model with 0% acc in around 1k Frames.    (b) LongVPO (Ours)

Figure 4: The V-NIAH results of our baseline InternVL2.5-8B and LongVPO. "Frame Depth" indicates the position where the needle image is located, ranging from 0% to 100% (from the beginning to the end of the video).

## 4.4 Ablation Study

### 4.4.1 Data Preparation

**Scene filtering in stage 1.** As shown in Tab. 2, we conduct ablations on scene filtering and response selection. In Stage 1, removing the scene filter—particularly when relying on simple Top-K selection—results in performance degradation. This underscores the importance of semantic filtering for robust long video understanding.

**Chosen response in stage 2.** In Stage 2, we compare three approaches for generating the chosen response: the target VLM directly processing the original video to produce responses (self-generated), responses generated by Qwen2.5-32B as used in our long-text context transfer method (Qwen-32B selected), and the target VLM receiving a combination of video frames and synthetic captions as input (interleaved). Surprisingly, the latter two methods yield suboptimal performance, highlighting the efficiency of our training process with only scene-question relevance labels.

**LLM backbone in stage 2.** LongVPO maintains strong performance even when using the 7B parameter InternLM2.5 as its backbone instead of Qwen2.5-32B, with only a slight drop observed. This demonstrates that its effectiveness is not dependent on a larger LLM.

### 4.4.2 Baseline Comparison

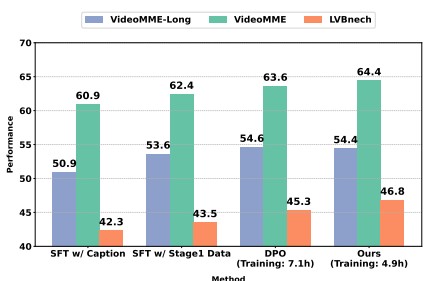

Figure 5: Comparison of Stage 1 training using SFT and DPO. Additional results are provided in the appendix.

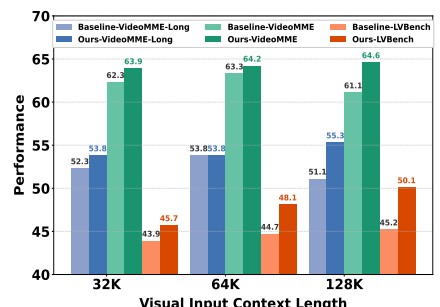

Figure 6: Performance scaling of LongVPO and InternVL2.5-8B with respect to increasing input frame counts.

**Scaling with input frames.** As the number of input frames increases, LongVPO exhibits progressively larger performance gains, whereas the baseline model (InternVL-2.5-8B) shows signs of saturation. As shown in Fig. 6, LongVPO achieves superior performance across benchmarks, in contrast to the baseline, which stagnates with longer contexts. These results confirm that LongVPO can more effectively extract and utilize temporal information from extended video sequences, highlighting its advantage in long-video understanding.

**Stage 1 training strategy.** As illustrated in Fig. 5, fine-tuning with single-video caption data leads to a noticeable performance drop. We attribute this to overfitting, since these captions were already part of the SFT dataset. Notably, conducting SFT on our synthesized Stage 1 data reverses this trend and achieves performance gains, validating the efficacy of our data construction pipeline. Furthermore,

Table 2: Ablation study on scene filtering and response selection methods across all long-video benchmarks. MLVU, LongVideoBench, and LVBench use a 32K context window during inference.

| Stage | Setting | MLVU | LongVideoBench | LVBench |
|---|---|---|---|---|
| Stage 1 | w/ Scene Filter | **72.9** | **66.1** | **45.3** |
| | w/o Scene Filter (adding a similar one) | 69.8 | 64.2 | 43.4 |
| | w/o Scene Filter (TopK) | 69.9 | 58.4 | – |
| Stage 2 | *Chosen response Choice* | | | |
| | Self-generated response | 72.9 | **66.1** | **45.3** |
| | LLM-generated (Qwen2.5-32B) response | **73.1** | 65.6 | 44.4 |
| | Self-generated w/ scene-interleaved caption | 73.0 | 66.1 | 44.7 |
| | *Long Context Knowledge Transfer Backbone* | | | |
| | InternLM2.5-7B instead of Qwen2.5-32B | 72.5 | 65.8 | 44.9 |

applying DPO using these preference pairs yields the most significant improvements, suggesting that the DPO framework enables more data-efficient learning and mitigates overfitting. Building on these findings to optimize DPO for long videos, we observe that, consistent with Fig. 1, using complete long videos as input to the reference model leads to suboptimal outcomes. Our ablation study confirms that this approach is less effective than our proposed method on both general and long-video benchmarks, while our method requires only approximately 70% of its training time.

### 4.4.3 Qualitative Comparison

Fig. 7 evaluates long video understanding through a pumpkin-carving action counting task. This challenging benchmark requires both action recognition and temporal instance tracking across extended durations. Current strong multimodal models (Qwen2.5-VL, Qwen2-VL, LLaVA-Video) failed to provide correct counts, while our LongVPO accurately identified all 5 instances. This demonstrates LongVPO's superior temporal understanding and counting capability in long videos, outperforming otherwise powerful baselines in complex comprehension tasks.

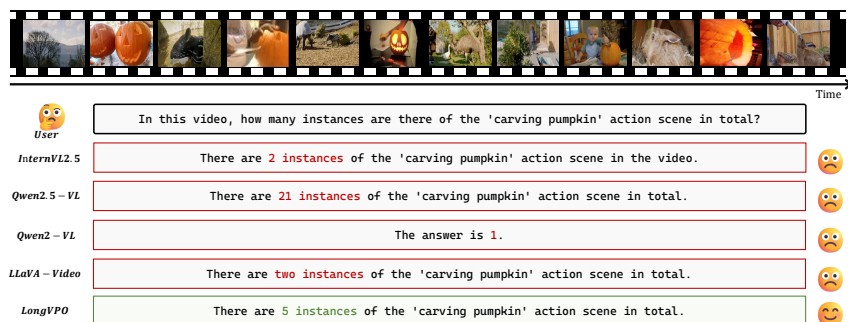

Figure 7: Qualitative comparison on long video understanding. More details are in the appendix.

## 5 Conclusion

We have proposed LongVPO, a novel two-stage DPO training framework tailored for long video understanding. By leveraging synthetic DPO instances constructed from short visual contexts, our method incrementally extends the capabilities of short-context VLMs to long-context comprehension, without relying on any annotated long-video data. Compared with specialized long-video models, LongVPO achieves the state-of-the-art on both long and short video understanding benchmarks. These advances highlight its potential as a general-purpose solution for long video understanding.

**Limitations.** Our work prioritizes performance improvement over inference computational efficiency. We will explore the integration with existing context compression approaches in future research.

## Acknowledgement

This work is supported by the National Key R&D Program of China (No. 2022ZD0160900), the Basic Research Program of Jiangsu (No. BK20250009), Nanjing University-China Mobile Communications Group Co., Ltd. Joint Institute (No. NJ20250033), and the Collaborative Innovation Center of Novel Software Technology and Industrialization.

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

## Appendix Overview

This appendix provides additional details of our approach and further experimental results, organized as follows:

- Section A describes the core experimental design behind our context-bias analysis.
- Section B showcases qualitative results across various scenarios.
- Section C outlines implementation details of our method.

| Model | LVBench | LongVideoBench | MLVU | Video-MME |
|---|---|---|---|---|
| Qwen2.5-VL | 45.3 | 56.0 | 70.2 | 65.1 / 71.6 |
| InternVL2.5 | 45.2 | 62.7 | 67.6 | 61.1 / 65.3 |
| + LongVPO | **50.1** ↑4.9 | **66.6** ↑3.9 | **74.1** ↑6.5 | 64.6 / 70.3 ↑3.5/↑5.0 |
| InternVideo2.5 | 47.4 | 63.2 | 72.8 | 63.3 / 71.1 |
| + LongVPO | **51.0** ↑3.6 | **67.2** ↑4.0 | **74.7** ↑1.9 | 66.1 / 73.1 ↑2.8/↑2.0 |
| Video-LLaMA3 | 45.3 | 59.8 | 73.0 | 66.2 / 70.3 |
| + LongVPO | **49.8** ↑4.5 | **63.4** ↑3.6 | **74.6** ↑1.6 | 67.2 / 71.4 ↑1.0/↑1.3 |

Table 3: Performance comparison on long video benchmarks. Improvements over base models are shown in red with ↑ symbols.

## A    Core Experimental Design for Context Position Bias Probing (Main Fig. 1)

**Evaluation Setup.** We directly selected tasks from MVBench [20] for evaluation. Only unambiguous tasks were included to ensure the validity of the labels.

To evaluate models designed primarily for short-context input, we introduce a simplified evaluation framework: (1) **Padding Strategy:** We simulate long-context scenarios by embedding the original video frame sequence into a larger grid (akin to high-resolution image tiling), surrounded by meaningless padding frames. (2) **Random Placement:** The original frames are randomly placed within this padded grid to test whether a model's performance is sensitive to the spatial location of meaningful content relative to the padding.

This experiment aims to validate two key constraints for an ideal long-video context model: (1) **Consistency across Context Lengths:** A well-designed model should maintain consistent performance across both short and long contexts without altering the task semantics. (2) **Position Invariance:** Since padding frames carry no meaningful information, the spatial location of valid video frames within the padded grid should not affect task performance.

As shown in Main Fig. 1, existing long-context models fall short of these expectations: (a) **Shifted Long-Context Consistency:** Performance varies with the distance between query and relevant frames, showing an undesirable sensitivity to position. An ideal long-video model should attend equally to relevant frames regardless of their location in the input. (b) **Short-Long Context Discrepancy:** Compared to our LongVPO, existing models exhibit a significant performance drop when transitioning from short to long context inputs. This suggests unreliable long-video understanding. When used directly as the reference model in DPO-style fine-tuning with long-context inputs, these models may lead to suboptimal performance. In contrast, LongVPO maintains nearly identical performance across short and long contexts, validating its robust design.

## B    Qualitative Results

We present additional qualitative comparisons with state-of-the-art models on long-video tasks across diverse scenarios. Despite being trained on synthetic data, our model demonstrates competitive open-ended QA performance, maintaining robustness across various domains.

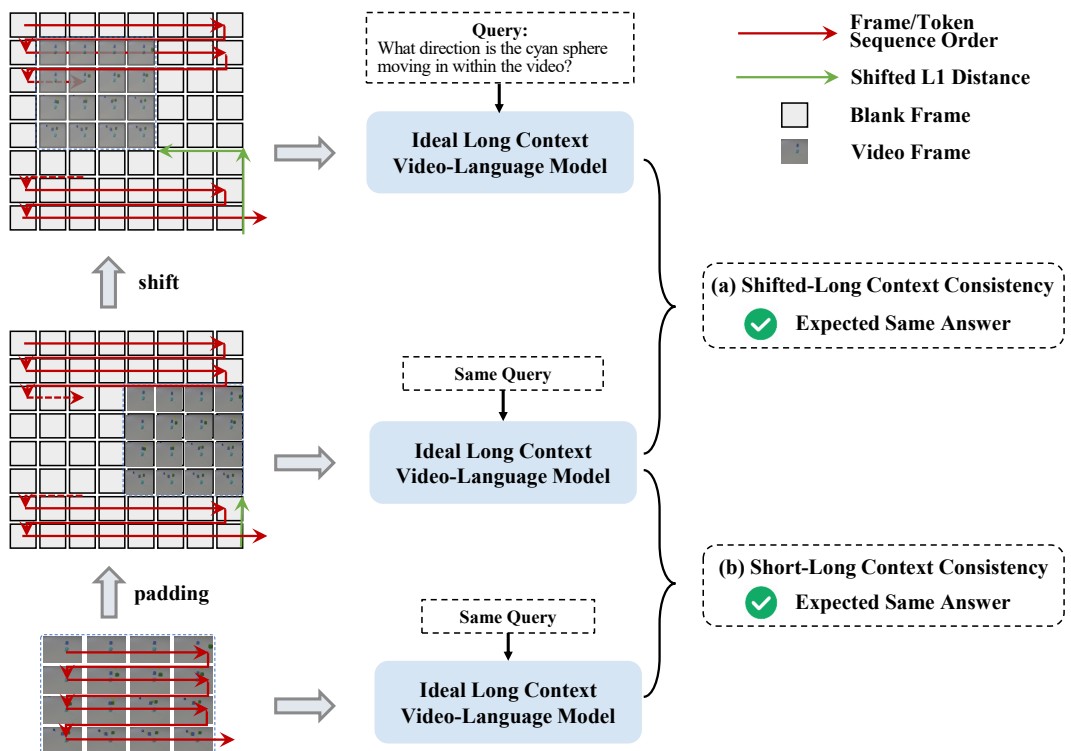

Figure 8: Core experimental design underpinning the context-bias analysis in Fig. 1.

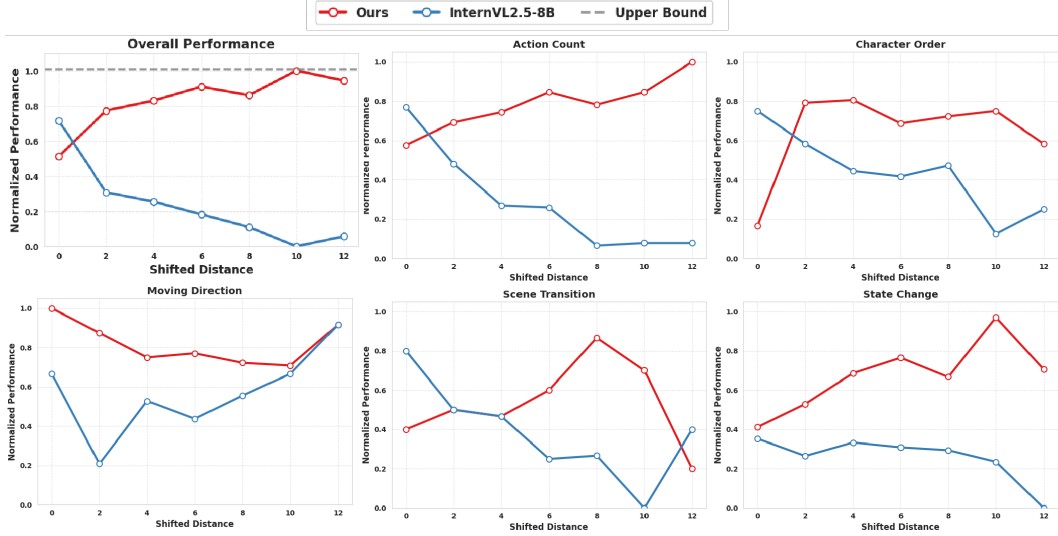

Figure 9: Compared to the main Fig. 1, we shorten the input video context length (by padding blank frames to a $10 \times 10$ grid rather than $12 \times 12$), yet the same "lost in the middle" phenomenon persists.

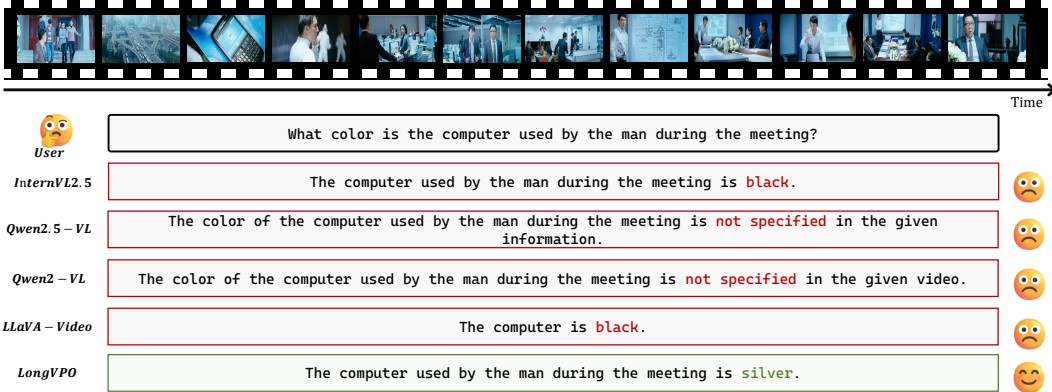

Figure 10: Long Video Understanding: Visual Semantic Understanding

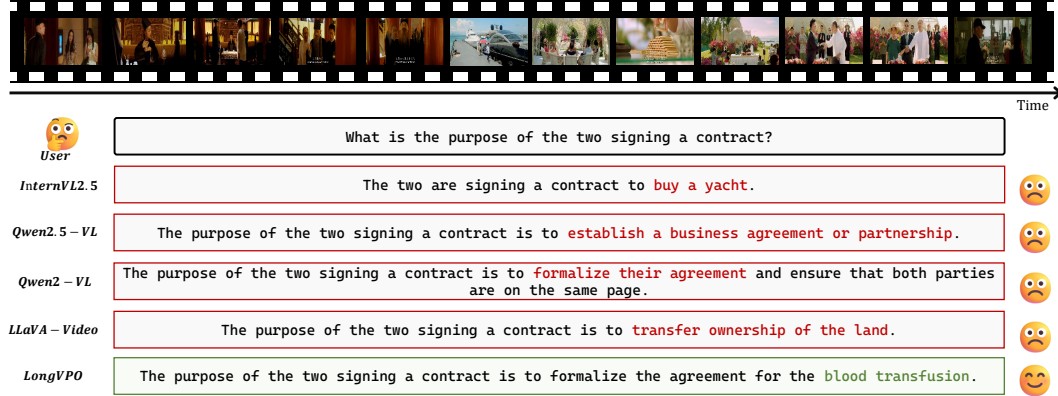

Figure 11: Long Video Understanding: Cross-temporal Scene Association

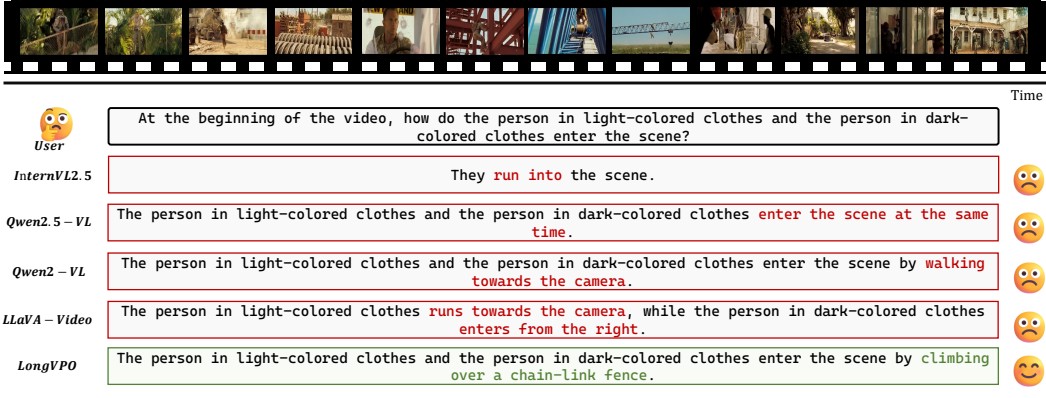

Figure 12: Long Video Understanding: Temporal Order Analysis

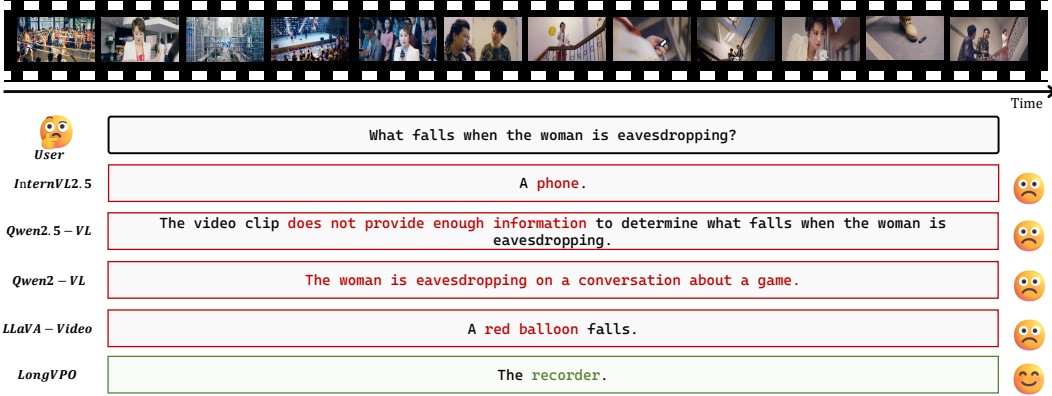

Figure 13: Long Video Understanding: Detail Comprehension

# C Implementation Details

**Hardware Setup.** All experiments were conducted on a server equipped with $4 \times 8$ NVIDIA H100 GPUs, each with 80GB of memory. We implement DeepSpeed Ulysses sequence parallelism to enable memory-efficient training.

**Training Strategy.** We adopt the proposed LongVPO method for training, which involves a two-stage fine-tuning process on a curated dataset of 16k samples—10k samples for stage 1 and 6k for stage 2. Each model variant is trained for 1 epoch, balancing training efficiency with the need for robust adaptation.

**Full-Model Fine-Tuning.** We fine-tune the entire model end-to-end. This includes the vision encoder, the vision-language connector, and the LLM backbone.

**Optimization Settings.** We use a composite loss that balances KL-divergence and supervised fine-tuning (SFT) objectives, with both weights set to 1.0 ($\beta$=0.01, $\alpha$=1.0). The learning rate is set to 5e-7, with batch size 8, a cosine learning rate scheduler, and a warm-up ratio of 0.01 to stabilize early training dynamics.

**Training Duration.** Each model variant requires approximately 10 hours of training with DeepSpeed Ulysses sequence parallelism enabled; otherwise, training completes in about 1 hour under the aforementioned configuration.

**Evaluation Settings.** For consistency and comparability, we follow the evaluation protocols established by InternVL2.5 and InternVideo2.5. The maximum number of frames per input is set to 512 to test the model's scalability in long-context understanding.

