# OpenReview forum: "LongVPO: From Anchored Cues to Self-Reasoning for Long-Form Video Preference Optimization"
_NeurIPS.cc/2025/Conference — NeurIPS 2025 poster_

### Official Review · Reviewer_vyqf · 2025-07-01

**Clarity:** 3
**Significance:** 2
**Originality:** 3
**Rating:** 4
**Confidence:** 4

**Summary:**

The paper tackles long-form video understanding without relying on expensive long-video annotations. It introduces LongVPO, a two-stage Direct Preference Optimization (DPO) pipeline. Using 16k synthetic preference pairs and no human labels, LongVPO outperforms previous open-source models on four long-video benchmarks.

**Questions:**

In Tab. 1 you compare LongVPO‐Stage 1/2 with other long-video VLMs that are not re-trained on your 16K synthetic triples. Could you  fine-tune the public InternVL-2.5-8B baseline on exactly your Stage 1 dataset under the same compute budget without any of your Stage 1/2 data

**Ethical Concerns:**

["NO or VERY MINOR ethics concerns only"]

**Final Justification:**

Thank the authors for the response. My concerns have been sovled and I will consider to improve my rating score.

**Limitations:**

Please see weakness

**Quality:**

3

**Strengths And Weaknesses:**

S1: Achieves strong long-video gains with only 16 k synthetic triples and no new human labels.

S2: Stage 1 addresses position bias; Stage 2 adds multi-segment reasoning, mirroring long-text LLM alignment.

S3: Introduces anchor-centric synthetic triples and applies both visual-similarity and question-specific filtering to keep supervision unique. The anchor-only approximation lets the reference model score just the anchor clip, avoiding long-sequence degradation while cutting compute.

W1: 	Competing long-video models are evaluated with their released weights; none are fine-tuned on the authors’ 16k dataset, so part of the gain may stem from extra training rather than technique.

W2: More ablations needed including fine-tune InternVL-2.5-8B with your Stage 1 data without DPO to isolate DPO’s contribution

W3: In Stage 2, “preferred” answers are self-generated and then reinforced via DPO, risking self-reinforcement bias. Consider adding independent judges (human or a separate VLM) or at least reporting results against ground-truth QA sets.

---

> ### Author Rebuttal · Authors · 2025-07-31
>
> **Q1: Comparison of fairness with released weights**
>
> We would like to clarify that our Stage 1 and Stage 2 training involved no additional manual annotations. All training signals are derived from the LLaVA-video-178k Caption dataset, **which is publicly available and already used as part of the InternVL-2.5-8B SFT training data [1].**
>
> **As suggested by reviewer dFSN, we report additional baseline results, including Video-LLaMA3 and InternVideo2.5, both of which have also been fine-tuned on more diversified video datasets that are either the same as or highly overlapping with LLaVA-video-178k.** Therefore, our method does not benefit from any original dataset.
>
> Our performance gains stem not from access to additional data, but rather from our method's capability to more efficiently organize and utilize existing data, through multi-stage optimization and explicit long-video reasoning.
>
> **Performance Summary**
> | Model          | LVBench     | LongVideoBench | MLVU        | Video-MME                 |
> | -------------- | ----------- | -------------- | ----------- | ------------------------- |
> | InternVL2.5   | 44.7        | 62.7           | 67.6        | 63.3 / 66.9               |
> | + LongVPO      | **50.1** (+5.4) | **66.6** (+3.9)    | **74.1** (+6.5) | **64.6** (+1.3) / **70.3** (+3.4) |
> | InternVideo2.5 | 46.4        | 60.6           | 72.8        | 63.3 / 71.1               |
> | + LongVPO      | **52.4** (+6.0) | **67.9** (+7.3)    | **74.5** (+1.7)       | **66.1** (+2.8) / **73.0** (+1.9)              |
> | Video-LLaMA3   | 45.3        | 59.8           | 73.0        | 66.2 / 70.3               |
> |+ LongVPO	| **49.8** (+4.5)	| **63.4** (+3.6)	| **74.6** (+1.6)	| **66.7** (+0.5) / **71.4** (+1.1) |
>
> **Q2: Isolating the contribution of DPO**
>
> As shown in Figure 5 of our main paper, we conducted experiments that fine-tuned InternVL-2.5-8B using only our Stage 1 caption data. Interestingly, we observed significant performance degradation, especially due to overfitting. This is because the Stage 1 Caption data overlaps with InternVL’s own SFT data, and repeated fine-tuning on this data leads to overfitting.
>
> Ablation Results (InternVL family):
>
> | Model                             | LVBench     | MLVU        | Video-MME                 |
> | --------------------------------- | ----------- | ----------- | ------------------------- |
> | InternVL2.5                          | 44.7        | 67.6        | 63.3 / 66.9               |
> | InternVL2.5-LongVPO-Stage1           | **46.8** (+2.1) | **72.9** (+5.3) | **64.2** (+0.9) / **70.1** (+3.2) |
> | InternVL2.5-SFT-Stage1-Based-Caption | 42.3        | 68.6        | 60.9 / 65.5               |
> | InternVL2.5-SFT-Stage1-Data          | 43.5        | 71.3        | 62.4 / 68.0               |
>
> We applied a second-stage SFT using preferred responses generated from our pipeline, which alleviates overfitting and demonstrates the effectiveness of our multi-stage design. These results suggest that while data quality matters, our training strategy and pipeline design are key contributors to performance improvement.
>
> **Q3: Risk of self-reinforcement bias in Stage 2**
>
> We understand the concern regarding self-reinforcement bias. However, under the context of limited long-video QA data, our focus is on developing a scalable, low-cost strategy to expand long-context capabilities in VLMs without relying on expensive human annotations.
>
> **1. The preference order is valid in most cases.** In Stage 2, we generate the preferred answer $y_0$ using the full video context, and construct a dispreferred answer $y_1$ using either an irrelevant scene or only a partial relevant one. Although the generated answer $y_0$ is not perfect, it tends to hallucinate less than $y_1$, which lacks the necessary context. Therefore, we argue that the preference order
> $\( y_0 \succ y_1 \)$  holds true in the majority of cases.
>
> **2. Empirical evidence against reinforcement bias.** To examine whether self-reinforcement causes overfitting or degradation, we repeated Stage 2 DPO training twice iteratively using InternVideo2.5. We observed consistent performance improvement across multiple benchmarks, indicating robust generalization rather than overfitting to earlier outputs.
>
> | Model              | LVBench | LongVideoBench | MLVU | Video-MME   |
> | ------------------ | ------- | -------------- | ---- | ----------- |
> | InternVideo2.5     | 46.4    | 60.6           | 72.8 | 63.3 / 71.1 |
> | + Stage 2 (iter 1) | 51.0    | 67.2           | 74.1 | 65.6 / 72.5 |
> | + Stage 2 (iter 2) | 52.4    | 67.9           | 74.5 | 66.1 / 73.0 |
>
> This result suggests that LongVPO has generalizable benefits, even under repeated preference sampling and DPO training.
>
> **3. On the use of external judges or human evaluation.** We agree that incorporating independent judges (e.g., Gemini 2.5 Pro or human annotators) could improve reliability. **However, such strategies would significantly increase the cost and conflict with the paper’s core goal of efficient and scalable training for long video reasoning.**
>
> Our approach aligns with the philosophy adopted by recent image-domain methods, such as MPO [1], which automatically constructed a large-scale preference dataset of over 1M pairs—mostly without the use of external judges. Although this scale is significantly larger than ours, MPO has demonstrated the effectiveness and scalability of self-training strategies on image reasoning tasks. **This supports the feasibility of similar approaches (i.e., self-training) in the long-video domain, especially under limited annotation resources.**
>
> Thus, while more rigorous preference curation could further improve results, we believe our current results are sufficient to demonstrate the promise of this self-training paradigm in the long-video setting. We plan to apply further preference filtering using LLMs and captions before open-sourcing our data, and leave more fine-grained supervision or human/VLM evaluation as future work.
>
> **References**
>
> [1] InternVL2.5 Technical Report: Expanding Performance Boundaries of Open-Source Multimodal Models with Model, Data, and Test-Time Scaling
>
> [2] Enhancing the Reasoning Ability of Multimodal Large Language Models  via Mixed Preference Optimization

---

> ### Comment · Reviewer_vyqf · 2025-08-06
>
> Thank the authors for the response. My concerns have been sovled and I will consider to improve my rating score.

---

> > ### Author Response · Authors · 2025-08-06
> > **Thank You for Your Constructive Feedback — We Will Improve the Final Version**
> >
> > Thank you very much for your thoughtful response and valuable suggestions. We really appreciate your detailed feedback and the time you took to engage with our work. Your comments have helped us clarify key points and improve the paper.
> >
> > We will carefully incorporate your suggestions into the final version to make the presentation clearer and more thorough.

---

### Official Review · Reviewer_dFSN · 2025-07-02

**Clarity:** 3
**Significance:** 3
**Originality:** 3
**Rating:** 4
**Confidence:** 4

**Summary:**

This paper propose LongVPO, a two-stage Direct Preference Optimization framework that enables short-context vision-language models to understand ultra-long videos without long-video annotations. In Stage 1, it generates synthetic preference data using short clips and filtered distractors to train the model efficiently. In Stage 2, it creates reasoning tasks from real long videos using recursive captioning and large language models for self-supervised fine-tuning. With only 16K synthetic examples and no human labels, LongVPO achieves state-of-the-art performance on multiple long-video benchmarks while retaining strong short-video understanding.

**Questions:**

1. Please seriously address the concerns raised in the Weaknesses section regarding the motivation and results. As a suggestion, validating the effectiveness of LongVPO on models with longer context VLMs (e.g., Qwen2.5-VL) may help alleviate these concerns.
2. DPO often causes the policy model to significantly deviate from the reference model. How does LongVPO address this issue, and are there any strategies employed to mitigate such deviations?

**Ethical Concerns:**

["NO or VERY MINOR ethics concerns only"]

**Final Justification:**

Given the excellent new results presented by the authors, as well as their contributions in long-video data construction and the DPO algorithm, I still think this paper is above the bar.

**Limitations:**

Please refer to the Strengths And Weaknesses

**Quality:**

2

**Strengths And Weaknesses:**

Strengths:
1. The paper is well-written with clear and coherent expression.
2. Half of the paper's motivation—how to efficiently generate high-quality training prompts for long videos—is highly meaningful.
3. The data generation pipeline in the two-stage DPO framework appears promising and may serve as a potential solution to the scarcity of training corpora for long-video tasks.

Weaknesses:
1. The other half of the paper's motivation—how to extend short-context VLMs to long videos—seems questionable. Unlike a year ago, most current foundation VLMs (e.g., Qwen2.5-VL, Video-LLaMA3, Seed1.5-VL) already support long context lengths and typically process videos using fixed FPS inputs. On this basis, extending short-context VLMs is no longer a necessary direction, either now or in the foreseeable future.
2. On certain datasets, increasing the number of input frames leads to very limited improvements, or even significant performance drops. This undermines the overall motivation of the paper and should be discussed in detail. Moreover, constructing a weaker baseline and showing improvements over it carries limited value. The reported relative gains of LongVPO should be adjusted to discount this inflated baseline effect.

---

> ### Author Rebuttal · Authors · 2025-07-31
>
> **Q1. On the motivation of extending short-context VLMs**
>
> We would like to clarify that our goal is not simply to "extend" legacy models, but to address a more fundamental and persistent challenge: **how to efficiently align the powerful, pre-existing semantic capabilities of foundation models for robust long-video reasoning.**
>
> **Our central insight is not model-specific but applies broadly: "Transferring and aligning strong short-context capabilities into long-context reasoning can be more efficient than direct long-context tuning."**
>
> While it is true that many foundation VLMs (e.g., Qwen2.5-VL, Video-LLaMA3, Seed-VL) now claim to support long-context inputs, several observations challenge the effectiveness of such support:
>
> * The training data landscape exhibits a pronounced long-tail distribution in context length. The vast majority of available multimodal supervision—such as image task pairs or short video clips—consists of short-context data. In stark contrast, annotated long videos suitable for instruction tuning or reasoning tasks are extremely scarce, as they demand substantial computational resources and costly labeling efforts. As a result, existing models risk being insufficiently trained for genuine long-context understanding.
>
>
> * On long-video benchmarks such as LongVideoBench, Video-MME, or LVBench, many long-context VLMs experience performance degradation as input length increases, highlighting the gap between architectural support and actual long-range reasoning ability.
>
> | Model          | LVBench         | LongVideoBench  | MLVU            | Video-MME                     |
> | -------------- | --------------- | --------------- | --------------- | ----------------------------- |
> | Qwen2.5-VL     | 45.3            | 56.0            | 70.2            | 65.1 / 71.6                   |
> | InternVL2.5       | 44.7            | 62.7            | 67.6            | 63.3 / 66.9                   |
> | + LongVPO      | **50.1** (+5.4) | **66.6** (+3.9) | **74.1** (+6.5) | **64.6 / 70.3** (+1.3 / +3.4) |
> | InternVideo2.5 | 46.4            | 60.6            | 72.8            | 63.3 / 71.1                   |
> | + LongVPO      | **52.4** (+6.0) | **67.9** (+7.3) | **74.5** (+1.7)       | **66.1 / 73.0** (+2.8 / +1.9)              |
> | Video-LLaMA3   | 45.3            | 59.8            | 73.0            | 66.2 / 70.3                   |
> | + LongVPO      | **49.8** (+4.5) | **63.4** (+3.6) | **74.6** (+1.6) | **66.7 / 71.4** (+0.5 / +1.3)              |
>
> **LongVPO is a highly efficient and general alignment method for both short- and long-context models.**
>
> - **First**, InternVL2.5-LongVPO itself already achieves performance that surpasses or matches most dedicated long-video models.
> - **More crucially**, our approach remains highly effective even when applied to specialized video models such as Video-LLaMA3 and InternVideo 2.5, which have already been fine-tuned on valuable long-video data.
>   - This is particularly noteworthy because the LLaVA-Video-178k dataset, from which we generate our preference data, was already included in the initial supervised fine-tuning (SFT) of these models.
>   - The fact that LongVPO still delivers significant additional gains demonstrates that its strength comes from the **alignment method itself**, not merely from exposure to new data, highlighting its power as a truly **efficient and general alignment technique**.
>
>
> **Q2. On limited gains from increasing frame count.**
>
> There is no significant drop in performance with more frames in our baseline models. **The previously reported LVBench score of 72.0 for InternVL2.5 (64 frames) was a known error; we have corrected it to 43.2, in line with the results reported in Video-LLaMA3's technical report.**
>
> Increasing the number of input frames from ​​64 → 512​​ yields consistent improvements (+0.6% avg). When combined with our ​​Stage 1​​ and ​​Stage 2​​ training, performance improves substantially, **absolute gain over the baseline: ​​+5.2%​​ (64frames), +4.6% (512frames).**
> | Model Variant     | LVBench  | LongVideoBench | MLVU     | Video-MME       | AVG\*      |
> | ----------------- | -------- | -------------- | -------- | --------------- | -------- |
> | InternVL2.5 (64)     | 43.2 (corrected)   | 60.0           | 68.9     | 64.2 / 66.9     | 59.4     |
> | InternVL2.5 (512)    | 44.7     | 62.7           | 67.6     | 63.3 / 66.9     | 60.0     |
> | + LongVPO Stage 1         | 46.8     | 66.1           | 72.9     | 64.2 / 70.1     | 63.2     |
> | + LongVPO Stage 2 | **50.1** | **66.6**       | **74.1** | **64.6 / 70.3** | **64.6** |
>
> *The average of two Video-MME scores is used to calculate the global average.
>
> **Q3. On DPO deviation from the reference model**
>
> We adopt a widely-used solution by Joint optimization with SFT loss (i.e.,  $L_{\text{total}} = L_{\text{LongVPO}} + L_{\text{SFT}}$  ) as common practice in many recent VLMs (e.g., [1]), ensuring that the model remains close to the reference distribution while still learning preference-based alignment to preserve generation quality.
>
> **Furthermore, we argue that some deviation from the reference model is not only acceptable, but desirable in our setting as an open research question**: Our goal is to equip the model with new capabilities (long-context reasoning) beyond the reference. Recent RLHF frameworks (e.g., DAPO [2], GLM-4.1V [3]) have removed KL-regularization between the policy model and the reference model to allow more flexible policy exploration.
>
> **References**
>
> [1] Enhancing the Reasoning Ability of Multimodal Large Language Models  via Mixed Preference Optimization
>
> [2] DAPO: An Open-Source LLM Reinforcement Learning System at Scale
>
> [3] GLM-4.1V-Thinking

---

> > ### Author Response · Authors · 2025-08-06
> > **Re: Submission 4855 Author Response**
> >
> > Dear Reviewer dFSN,
> >
> > Thank you again for your valuable feedback and supportive comments on our work!
> >
> > We have made effort to address your concerns in our previous response. As the author-reviewer discussion period is only 3 days left, we would greatly appreciate it if you could let us know whether our reply sufficiently resolves your questions.
> >
> > If you have any further comments or concerns, we would be more than happy to discuss them!

---

> > ### Comment · Reviewer_dFSN · 2025-08-06
> > **Official Comment by Reviewer dFSN**
> >
> > Thank the authors for the response. The reply has addressed some of my concerns. However, regarding the motivation, I remain unconvinced: the claim that long-context VLMs experience performance degradation as input length increases may simply be due to the gap between training and testing. As for why training does not support the longest possible input lengths, this touches on a more fundamental question — is longer always better? Do the gains from supporting longer contexts justify the significantly increased training costs? In fact, models like the InternVL series and LLaVA-Video have already achieved impressive performance on long video understanding tasks using no more than 64 frames, which might be the more ideal approach. That said, given the excellent new results presented by the authors, as well as their contributions in long-video data construction and the DPO algorithm, I still think this paper is above the bar.

---

> ### Author Response · Authors · 2025-08-07
>
> Thank you for your constructive feedback. We understand your concerns regarding the motivation behind long-context VLMs. While the train-test gap may contribute to performance degradation, **the intrinsic limitations of sparse sampling methods** present a more fundamental challenge, especially for long-video understanding. We would like to discuss your points below:
>
> **1. On "Longer is Not Always Better"**
>
> Current benchmarks reveal critical limitations of fixed-length sparse sampling (e.g., 64 frames):
>
> - **Human-level understanding requires dense context**:  According to the current benchmark for advanced video reasoning and understanding ([1], Fig 9), human performance shows a significant improvement with increasing the number of frames from 1 to 256, reaching 90% accuracy, while GPT-4o/LLaVA-Video plateau at around 8 frames with performance below 50%, indicating a ~40% performance gap.
>
> - **Significant performance drop** occurs as video duration increases *under identical frame budgets*:
>   - LLaVA-Video-72B on VideoMME: Short[< 2min] (81.4%) vs. Long[30min ~ 60min] (61.5%); on LongVideoBench [8-15s] (72.4%) vs. [900-3600s] (59.3%).
>   - Contrary to the performance claims in Our Main Figure 4, Sparse sampling ​​fundamentally cannot achieve reliable needle-in-a-haystack retrieval​​ [2].
>   - This huge gap suggests sparse sampling struggles to robustly represent temporal sparse information in long videos.
>
> - **Why Sparse Sampling Fails for Precise Queries: LVBench** (adopted by Gemini in their technical report [3] to demonstrate long-video capabilities)
>
>    + *Question: "How does the goalkeeper prevent Liverpool's shot at 81:38 in this 1:40:20 match?"*
>
>    + Sparse sampling struggles to reliably locate such precise moments without luck. In contrast, long-context models are specifically designed for systematically processing relevant segments, ensuring more accurate results.
>
> **2. So, Do We Truly Need Native Long-Context Support?**
>
> While key-frame extraction (e.g., [4]) can mitigate sparse sampling issues, it introduces ​​pipeline complexity​​ and ​​information loss​​ (disrupting temporal continuity). In contrast, ​​native long-context modeling​​ enables:
>   - **Complex Interaction**: Real-world applications (e.g., multi-turn agent operation interaction, streaming chatting) require retaining and reasoning over linearly growing video-history contexts.
>   - **Reasoning enhancement**: Recent work [5] demonstrates that long-context capabilities are essential for advanced reasoning, and we believe this also applies to the VLM domain.
>
> **3. Training Cost Efficiency**
>
> Our approach achieves competitive results with *modest resource investment* compared to large scale long context SFT:
> - The total training time is ～2 hours on 4×8 H100 GPUs for Internvideo2.5-7B and <10 hours on 4×8 H100 GPUs for Internvl2.5-7B.
> - Stage-1 training cost reduced to 70% through LongVPO optimization, as concerned by Reviewer 4ZfK.
>
> Native long-context support not only aims to bridges the performance gap but also lays the foundation for complex video tasks. We believe these advances, coupled with our contributions to pipeline construction and efficient DPO training, justify the paper's significance.
>
> ---
>
> **References**
> [1] Towards Video Thinking Test: A Holistic Benchmark for Advanced Video Reasoning and Understanding
>
> [2] Long Context Transfer from Language to Vision
>
> [3] Advancing the frontier of video understanding with Gemini 2.5
>
> [4] VideoTree: Adaptive Tree-based Video Representation for LLM Reasoning on Long Videos
>
> [5] Longer Context, Deeper Thinking: Uncovering the Role of Long-Context Ability in Reasoning

---

### Official Review · Reviewer_4ZfK · 2025-07-02

**Clarity:** 3
**Significance:** 3
**Originality:** 3
**Rating:** 4
**Confidence:** 4

**Summary:**

This paper proposes a fine-tuning method derived from direct preference optimization (DPO) for long-form video understanding. The authors observed two limitations of existing video language models. With a simple padding, the authors empirically show that the video-language models have positional bias in Figure 1. In addition, in Figure 4, the authors analyzed the performance degradation of existing video-language models by evaluating performance across various lengths of videos. As the video gets longer, the baseline models show significant performance loss. The authors tackled the problem without sacrificing the short-video performance via a two-stage post-training method. First stage utilizes anchored cues and modified DPO loss with them. And in the second stage, self-training was applied. Overall, the proposed method provides considerable performance gain on multiple video benchmarks, including LVBench, LongVideoBench, MLVU,  Video-MME, and MVBench.

**Questions:**

1. Why better robustness is important to understanding long-form videos? The figure 1 explains one of the main observations that the baseline methods exhibit a significant degradation when the location of the main content changes. It is not clear why tackling this problem improves long video understanding.

2. Did the authors observe any artifacts or disadvantages of fine-tuning with synthetic or artificially derived data?

**Ethical Concerns:**

["NO or VERY MINOR ethics concerns only"]

**Final Justification:**

The proposed method achieves a significant performance gain compared to various methods, including one of the most popular alignment method, Direct DPO. Also, it improves efficiency as well. Lastly, the author response addresses the major concerns originally raised.

**Limitations:**

Limitation of this work was discussed by the authors with two lines.

**Quality:**

3

**Strengths And Weaknesses:**

**Strengths.**

1. **Presentation.** This paper is well-organized and reads well. This paper started from an interesting observation that the video models have positional bias. Also, fine-tuning for long-form videos leads to degradation in short-form video performance. The motivations/observation in Figure 1. ad Figure 4.and the overall pipeline in Figure 2 are helpful to understand the main argument.
2. **Motivation.** This a bit sounds redundant but the presentation was good but also the observations themselves are interesting. In many studies, people think that the longer video understanding is challenging and improving longer video understanding is the only focus. But improving long and short video understanding both is quite interesting.
3. **Interesting trick.** Anchor-only approximation and the DPO objective with the anchor is a simple and effective trick.Presumably, this may accelerate fine-tuning since the inference of reference model gets faster. Also, the reference model (probably a existing model for short-form videos at the beginning) will provide a better reference probability.

**Weaknesses.**

1. **Analysis for approximation error.** Although the approximation with anchored cues is an interesting technique, the approximation gap is not fully analyzed. The experiment to check whether the approximation provides a performance boost or degradation.
2. **Incomplete comparison.** The main table, table 1 has many missing values. Especially, InternVL2.5 with 64 frames looks strong. Also, it is comparable or more competitive than InternVL2.5 with 512 frames. Unfortunately, some entries for InternVL2.5 with 64 frames are missing which outperforms on average compared to InternVL2.5 with 512 frames. If the table has all entries, then the effectiveness of the proposed method can be evaluated more accurately.
3. **No computational complexity comparison.**  The proposed method is a two-stage approach. It requires new tricks. Although the proposed method provides the performance boost, without the comparison regarding computational cost, it is hard to precisely evaluate the value of the proposed method.

---

> ### Author Rebuttal · Authors · 2025-07-31
>
> **Q1: Stage 1 Approximation Error Analysis**
>
> To directly analyze the “approximation gap,” we conducted a head-to-head comparison between Stage 1 LongVPO and a standard DPO baseline (reference model on the full synthetic video).
>
> The table below summarizes the results, including the original 128-frame/32k-token setting from Figure 5 of the main paper, and an extended 256-frame/64k-token experiment that further demonstrates scalability.
>
> **Performance Gains of LongVPO vs. Direct DPO (Stage 1)**
> | Max Frames / Visual Context | LVBench Δ | LongVideoBench Δ | Video-MME Δ | Training Cost    |
> | --------------------------- | --------- | ---------------- | ----------- | ---------------- |
> | 128 frames / 32k tokens     | +1.5%     | +0.1%            | +1.2%       | 4.9 h vs. 7.1 h (70%)  |
> | 256 frames / 64k tokens     | +2.0%     | +0.9%            | +1.0%       | 8.0 h vs. 11.5 h (70%) |
>
> Stage 1 approximation consistently improves performance, especially at longer contexts.
>
> **Q3: Computational Complexity**
>
> From the table in Q1:
> * Stage 1: Our LongVPO Stage 1 achieves better performance than standard DPO while requiring ~30% less training time, demonstrating its efficiency advantage.
> * Stage 2: The training cost is identical to standard DPO, ensuring that the overall improvement does not come at additional computational expense. Both cost 4.4h.
>
> **Q2: Incomplete Comparison (Table 1)**
>
> * **For InternVL 64Frames**, we corrected the missing/broken entries in our initial submission and fixed a known LVBench score error (72.0 → 43.2) for InternVL2.5 (64 frames), consistent with the reproduced results in Video-LLaMA3’s report. **The updated results confirm that the InternVL-2.5 (512 frames) model is the strongest baseline.**
>
> | Model Variant     | LVBench          | LongVideoBench | MLVU | Video-MME | MVBench | AVG\* |
> | ----------------- | ---------------- | -------------- | ---- | --------------- | ------- | ----- |
> | InternVL2.5 (64)  | 43.2 (corrected) | 60.0           | 68.9 | 64.2 / 66.9	           | 72.0    | 61.9  |
> | InternVL2.5 (512) | 44.7             | 62.7           | 67.6 | 	63.3 / 66.9           | 72.0    | 62.4  |
> | + LongVPO Stage 1 | 46.8             | 66.1           | 72.9 | 64.2 / 70.1           | 72.9    | 65.2  |
> | + LongVPO Stage 2 | 50.1             | 66.6           | 74.1 | 64.6 / 70.3           | 73.1    | 66.3  |
>
> *AVG is computed using the mean of the two Video-MME scores.
>
> * **Regarding the missing data for models other than InternVL**, all figures in the table are cited directly from their respective official reports. These entries are missing because the original authors did not publish results for these specific benchmarks.
>
> **Q4: Why Localizing Robustness Matters in Long-Form Video Understanding？**
>
> Robustly localizing key information and gathering information to answer correctly is not just a proxy task; it is a fundamental prerequisite for any meaningful long-video understanding as a Common solution paradigm [1][2]. The reasoning is twofold:
> * **Sparsity of Information**: User queries about long videos often depend on a few critical moments. The model must first reliably find this "needle" before it can reason about it.
> * **Prerequisite for Reasoning**: Complex reasoning requires integrating evidence, often from multiple, non-contiguous scenes. If the model cannot robustly locate each piece of evidence, its reasoning process fails at the first step.
>
> Our positional bias experiment in Figure 1 directly simulates this challenge. This is analogous to the widely used "needle-in-a-haystack" tests[3] in long-text evaluation, which are designed specifically to pressure-test a model's ability to retrieve sparse, critical information from a long, noisy context. Without this robustness, a model cannot truly comprehend long-form content.
>
> **Q5: Artifacts of Tuning with Synthetic Data**
>
> We did observe minor stylistic artifacts in a small subset of open-ended generation tasks (e.g., summarization). These typically involve the model adopting a structural narration style, using phrases like
> > "The video begins with..." or "The scene transitions to...".
>
> These artifacts are a byproduct of the captioning style used to generate training data and **do not harm factual accuracy on question-answering tasks (Please see the qualitative comparison in the main paper and appendix for more examples)**. We view this as a data diversity challenge that can be effectively addressed in future work with simple techniques like style-aware prompting, rather than a fundamental flaw in the model's reasoning capabilities.
>
> **References**
>
> [1] Deep Video Discovery: Agentic Search with Tool Use for Long-form Video Understanding
>
> [2] VideoTree: Adaptive Tree-based Video Representation for LLM Reasoning on Long Videos
>
> [3] Needle In A Haystack – Pressure Testing LLMs

---

> > ### Comment · Reviewer_4ZfK · 2025-08-05
> >
> > I thank the authors for the elaborated response with multiple additional experimental results/analyses. I am glad that the proposed method achieves a significant performance gain compared to Direct DPO with lower computational cost. The response addresses the major concerns.

---

> ### Author Response · Authors · 2025-08-06
> **We Appreciate Your Valuable Feedback and Will Enhance the Final version**
>
> Thank you for your valuable comments and insights, which allow us to better highlight the strengths of LongVPO. We will carefully take your suggestions to enhance the clarity and persuasiveness of the revised version.

---

### Official Review · Reviewer_FVvM · 2025-07-05

**Clarity:** 2
**Significance:** 3
**Originality:** 2
**Rating:** 4
**Confidence:** 3

**Summary:**

The paper introduces LongVPO, a new two-stage training framework designed to enhance the long-video understanding capabilities of Vision-Language Models (VLMs) without relying on costly human annotations for long videos. The core problem it addresses is that existing VLMs, typically trained on short clips, struggle with ultra-long videos due to a lack of relevant training data and inherent positional biases (a "lost-in-the-middle" effect). The authors demonstrate that LongVPO significantly outperforms existing open-source models on several long-video benchmarks (LVBench, LongVideoBench, MLVU, Video-MME) and maintains strong performance on short-video tasks (MVBench), all while using only 16K synthetic examples and no direct human labels for long videos.

**Questions:**

A critical piece of evidence is missing from the ablation study. To convincingly argue for the necessity of the reference model approximation in Stage 1 (a central claim), it seems essential to compare it against the most direct baseline: standard DPO training where the reference model is fed the entire interleaved clip sequence.

Could you please provide this comparison? If the standard DPO baseline performs significantly worse, it would strongly validate your proposed approximation as a critical and necessary innovation. Without this, the motivation for the approach remains incomplete.

**Ethical Concerns:**

["NO or VERY MINOR ethics concerns only"]

**Quality:**

3

**Strengths And Weaknesses:**

This paper presents an interesting and promising approach to the challenging problem of long-form video understanding. The proposed two-stage framework sounds interesting, and the empirical results show notable gains over existing open-source models.

Strengths
- The paper tackles a timely and highly relevant problem. Developing scalable methods for long-video understanding that do not rely on prohibitive annotation costs is a significant goal for the community.
- The proposed two-stage DPO framework seems interesting. The "anchored cues" concept in Stage 1, especially the pragmatic approximation of the reference model to handle long contexts, is a clever way to adapt DPO to this setting. The self-training loop in Stage 2 is an attempt at enabling models to generate their own complex training curricula.
- The model demonstrates strong performance, achieving state-of-the-art results on multiple long-video benchmarks compared to other open-source models. This suggests the proposed training strategy is effective.


Weaknesses / Major Concerns
- My primary concern is the heavy reliance on a very powerful external model (Qwen-2.5 32B) for the critical data generation step in Stage 2. It is difficult to disentangle the contribution of the LongVPO framework itself from the sheer capability of this external LLM to generate high-quality reasoning tasks. The paper presents itself as a self-contained method, but its success may be largely attributable to an external, unanalyzed component. This makes it hard to assess the generalizability and true effectiveness of the framework.
- The paper argues that approximating the DPO reference model in Stage 1 is a key contribution. However, it fails to include a direct comparison against a standard DPO baseline where the reference model processes the full synthetic video. Without this experiment, the claim that the approximation is necessary for performance (and not just efficiency) is unsubstantiated. This is a significant gap, as it leaves a core methodological claim partially unverified.

---

> ### Author Rebuttal · Authors · 2025-07-31
>
> **Q1: Concern Regarding the Role of Qwen-2.5 32B in LongVPO Stage2**
>
> We would like to clarify that LongVPO only uses Qwen-2.5 32B in a limited role: 1) It generates questions from video captions, and 2) provides binary relevance hints for each video segment. Most importantly, Qwen-2.5 32B is not used to generate the model’s preferred responses in Stage 2.
>
> To further address the reviewer’s concern, we conducted additional ablation studies:
>
> **1. LongVPO without Qwen-2.5 32B exhibits only slight performance degradation.** We replaced Qwen-2.5 32B with InternLM2.5-7B-chat (the backbone LLM of our baseline InternVL2.5-8B). This version, LongVPO-InternLM-Stage2, shows only marginal degradation:
>
> | Model                   | LVBench | LongVideoBench | MLVU | Video-MME |
> | ----------------------- | ------- | -------------- | ---- | --------- |
> | LongVPO-Stage2          | 50.1    | 66.6           | 74.1 | 64.6      |
> | LongVPO-InternLM-Stage2 | 49.7    | 66.2           | 73.7 | 64.4      |
>
> **2. Directly using Qwen2.5-32B Generated Preferred Responses is much worse.** As shown in Table 2 of the main paper: Directly using Qwen-2.5 32B responses as preferred answers for DPO performs worse than using our model’s self-generated responses: with LVBench: -1.7% and LongVideoBench: -1.9% This highlights that the external model does not dominate the learning signal and its deep involvement is not beneficial to performance.
>
> 3.The reliance on Qwen-2.5 32B is primarily for its instruction-following and formatting capabilities, e.g., outputting desired JSON-formatted question-answer pairs. **Fundamentally, our method focuses on enabling the VLM to self-correct its own response preferences based on valid guidance, rather than directly depending on a powerful external model—this is also the key distinction from SFT or other methods that rely heavily on strong external supervision.**
>
> **Q2: On the Necessity of Stage 1 Approximation vs. Direct DPO**
>
> * Figure 5 Recap: Our main paper’s Figure 5 shows that standard DPO (reference model on full synthetic video) underperforms LongVPO on Video-MME and LVBench, due to weaker long-context alignment.
>
> * Extended Ablation: We further increased the max video length from the original 128 frames / 32k visual tokens training input to 256 frames / 64k tokens. As training context length grows, LongVPO’s Stage 1 approximation consistently yields extra gains, confirming that our design is not only more efficient but also more effective than a naïve full-video DPO baseline.
>
> **Performance Gains of LongVPO vs. Direct DPO in Stage 1 Data**
> | Max Frames / Visual Context | LVBench Δ | LongVideoBench Δ | Video-MME Δ | Training Cost    |
> | --------------------------- | --------- | ---------------- | ----------- | ---------------- |
> | 128 frames / 32k tokens     | +1.5%     | +0.1%            | +1.2%       | 4.9 h vs. 7.1 h  |
> | 256 frames / 64k tokens     | +2.0%     | +0.9%            | +1.0%       | 8.0 h vs. 11.5 h |
>
>
>
> We hope these additional results sufficiently address both concerns, as there is only minimal impact of external LLM reliance and the effectiveness of our Stage 1 approximation beyond efficiency.

---

> ### Author Response · Authors · 2025-08-06
> **Re: Submission 4855 Author Response**
>
> Dear Reviewer FVvM,
>
> Thank you for your insightful feedback. We are writing to follow up on our recent rebuttal of our manuscript. We sincerely appreciate the time and effort you dedicated to your thoughtful review.
>
> With the discussion period ending in 3 days, we are keen to ensure we have resolved your concerns. Please let us know if our revisions are satisfactory or if anything else requires clarification.

---

### Decision · Program_Chairs · 2025-09-17

**Decision:**

Accept (poster)

**Comment:**

This paper proposes LongVPO, a two-stage Direct Preference Optimization framework that enables short-context vision-language models to understand ultra-long videos without long-video annotations.
The authors provide several experimental results that convince reviewers to raise their ratings. The final ratings are all borderline acceptance. The remaining concern is the unconvinced claim that long-context VLMs experience performance degradation as input length increases may simply be due to the gap between training and testing. However, most reviewers agree that the proposed method achieves a significant performance gain compared to Direct DPO with lower computational cost. Hence, the AC recommends acceptance.